# Explainable emphysema detection on chest radiographs with deep learning

**Erdi Çallı** *, **Keelin Murphy, Ernst T. Scholten, Steven Schalekamp** , **Bram van Ginneken**

Diagnostic Image Analysis Group, Radboudumc, Nijmegen, The Netherlands

* erdicalli@gmail.com

**Citation:** Çallı E, Murphy K, Scholten ET, Schalekamp S, van Ginneken B (2022) Explainable emphysema detection on chest radiographs with deep learning. PLoS ONE 17(7): e0267539. https://doi.org/10.1371/journal.pone.0267539

**Data Availability Statement:** The data is publicly available at {https://doi.org/10.5281/zenodo.6373392}.

**Funding:** The author(s) received no specific funding for this work.

## Abstract

We propose a deep learning system to automatically detect four explainable emphysema signs on frontal and lateral chest radiographs. Frontal and lateral chest radiographs from 3000 studies were retrospectively collected. Two radiologists annotated these with 4 radiological signs of pulmonary emphysema identified from the literature. A patient with ≥2 of these signs present is considered emphysema positive. Using separate deep learning systems for frontal and lateral images we predict the presence of each of the four visual signs and use these to determine emphysema positivity. The ROC and AUC results on a set of 422 held-out cases, labeled by both radiologists, are reported. Comparison with a black-box model which predicts emphysema without the use of explainable visual features is made on the annotations from both radiologists, as well as the subset that they agreed on. DeLong's test is used to compare with the black-box model ROC and McNemar's test to compare with radiologist performance. In 422 test cases, emphysema positivity was predicted with AUCs of 0.924 and 0.946 using the reference standard from each radiologist separately. Setting model sensitivity equivalent to that of the second radiologist, our model has a comparable specificity ($p = 0.880$ and $p = 0.143$ for each radiologist respectively). Our method is comparable with the black-box model with AUCs of 0.915 ($p = 0.407$) and 0.935 ($p = 0.291$), respectively. On the 370 cases where both radiologists agreed (53 positives), our model achieves an AUC of 0.981, again comparable to the black-box model AUC of 0.972 ($p = 0.289$). Our proposed method can predict emphysema positivity on chest radiographs as well as a radiologist or a comparable black-box method. It additionally produces labels for four visual signs to ensure the explainability of the result. The dataset is publicly available at https://doi.org/10.5281/zenodo.6373392.

## Introduction

Emphysema is a leading form of Chronic Obstructive Pulmonary Disease (COPD), which affects approximately 4.6% of the US population [1]. Chest radiograph (CXR) is typically the first and most common imaging examination for patients presenting with respiratory

**Competing interests:** The authors have declared that no competing interests exist.

symptoms. Especially with patients of COVID-19, emphysema detection is crucial in patient management, because it significantly increases the intensive care unit admission rates, increased respiratory support requirements and higher invasive mechanical ventilation frequencies [2]. This indicates the importance of detecting signs of emphysema in CXRs.

The automated diagnosis of emphysema on CXR has received relatively little attention to date. Coppini et al. (2007) [3], Coppini et al. (2013) [4], and Miniati et al. [5] used lung shapes to detect emphysema claiming performances of 0.90 accuracy, 0.954 area under the receiver operating curve (AUC), and 0.955 AUC respectively. These three studies use handcrafted features with neural networks on small datasets. More recently, in a small patient group of 80, Wanchaitanawong et al. (2021) [6] proposed that AI-based emphysema scores from CXRs could be used for patients who cannot perform spirometry and achieve similar results in diagnosing COPD.

Campo et al. (2018) [7] created chest radiograph projections from CT and achieved 0.907 AUC in predicting CT-based emphysema scores from these. Some studies used different modalities that look similar to a CXR or a derivative of the CXR to detect emphysema. For example, scout images taken as a part of the CT scanning process were used to evaluate emphysema severity in [8]. This study has shown that a deep learning method can predict the emphysema severity from scout images with results consistent with CT quantification. Darkfield radiographs were also used to predict emphysema in [9], using CT based scores as the reference standard. In that study an AUC of 0.79 was obtained in detecting mild emphysema in a study involving 83 patients.

There are up to 31 deep learning based studies using the ChestXray14 dataset [10], potentially providing an emphysema label among 13 others [11]. However, most of these studies use automatically extracted labels, which are noisy and unsuited to evaluation [12], including subcutaneous emphysema under the emphysema label for example [13]. The most recent such work [14] proposes an attention based extension of DenseNet121 [15] and achieves a 0.933 AUC in detecting emphysema, however the known issues with emphysema labeling in that dataset [13] makes this result difficult to interpret. Few studies collect radiologist annotations for various diseases and evaluate model performances on this data. Li et al. (2021) [16] uses annotations from 3 radiologists on 10,738 CXRs from hospital archives and reports an AUC of 0.942 on predicting emphysema as a part of the ChestXray14 disease labels. Lin et al. (2020) [17] collects localized features based on the ChestXray14 disease labels as well as 4 viral pneumonia labels for 310 CXRs, however does not report model performance on individual diseases.

In this work, we describe a deep learning system that is trained and evaluated on radiologist labeled frontal and lateral CXRs. It is designed to provide an explainable emphysema score including the prediction of four visual signs of emphysema defined in the literature [18]. Sutinen et al. [18] proposed that the flattening of the diaphragms in frontal and lateral CXR, irregular radiolucency on the frontal CXR, and an abnormally large retrosternal space in the lateral CXR are key signs for detecting emphysema. This was confirmed by Miniati et al. [19] using a group of 458 patients and 5 readers, achieving 90% sensitivity and 98% specificity. Images depicting these 4 visual signs are provided in Fig 1.

In addition to being accurate and reliable, a clinically relevant deep learning method needs to produce explainable results in order to gain trust and acceptance from end users. Although many studies provide visual cues on which pixels or locations are contributing most for a prediction [20–23], this information may not be aligned with the expertise of the radiologist [24, 25] and can potentially lead to confusing explanations, hindering acceptance of the method. Recently some studies have worked to create links between radiologically understood concepts

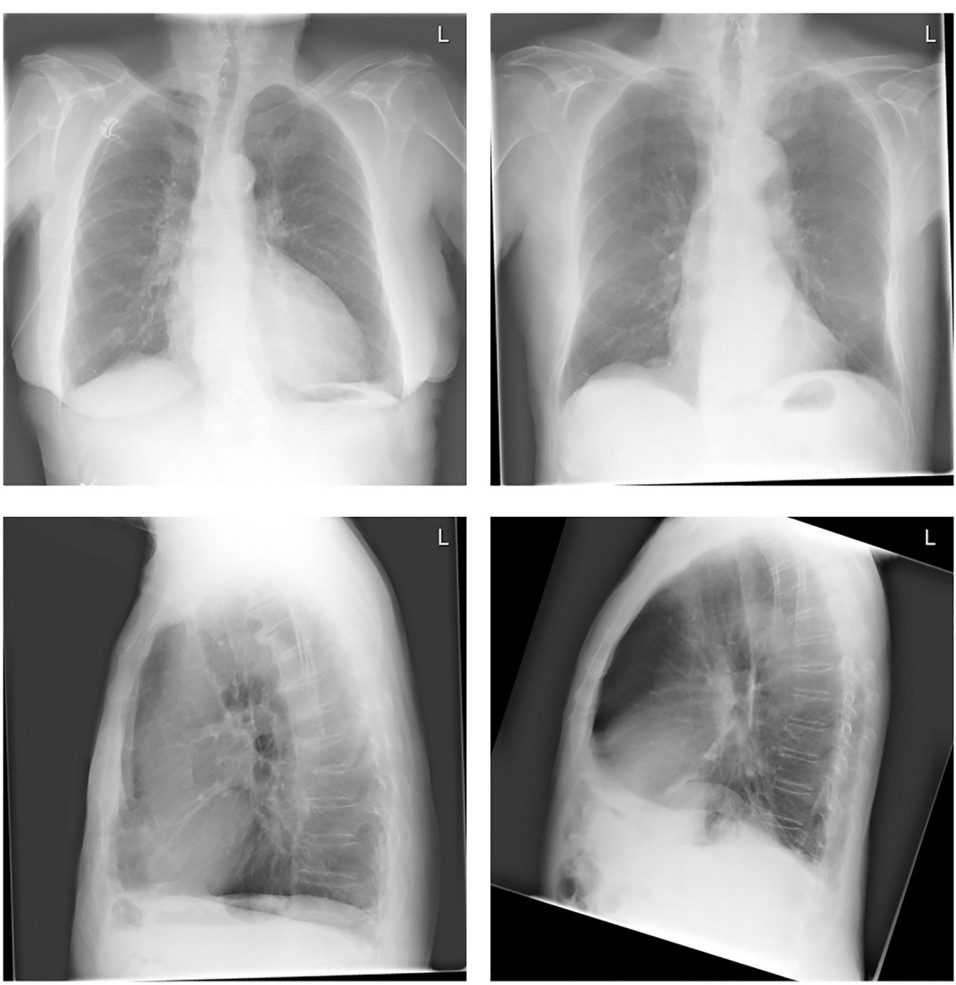

**Fig 1. The 4 radiological signs.** The top row shows the radiological signs on frontal chest radiographs. The left frontal chest radiograph shows the flattening of the diaphragm, and the right shows irregular radiolucency. The bottom two chest radiographs show the radiological signs on the lateral chest radiograph. The left lateral chest radiograph clearly shows the flattening of the diaphragmatic contours while the right demonstrates an abnormal retrosternal space.

and what a deep learning model predicts [26, 27]. In this work, we use labels from an established radiological protocol to predict emphysema, ensuring that end users can connect the outcome with their expert domain knowledge.

To build and evaluate our explainable deep learning system, we collected frontal and lateral chest radiographs for 3000 studies. These were annotated by two radiologists for the existence of each of the 4 described visual signs. The descriptions provided to the radiologists are included in Table 1. This annotated data was used to train and evaluate deep learning models for the prediction of each visual sign and a final emphysema score. Performance was compared with each of the two experienced radiologists and additionally with a black-box method which provides an emphysema label without explainable visual signs. We show that neither the radiologists nor the black-box method outperform the proposed explainable model in detecting emphysema. With 3000 studies, this is the largest study to date using the four visual signs to detect emphysema on CXR, and the first to use deep-learning for this task, providing radiologically explainable results.

**Table 1. The four signs of emphysema as described by Sutinen et al. [18].**

| Sign | Description |
| --- | --- |
| Frontal—Flattening of the diaphragm | Depression and flattening of the diaphragm with blunting of costophrenic angles. The actual level of the diaphragm is not as significant as the contour. The body build of the individual should also be considered. For example, in a short, stocky individual, emphysema might be diagnosable even if the diaphragm were at the level of the tenth rib posteriorly. |
| Frontal—Irregular radiolucency | Irregular radiolucency of the lung fields. This manifestation is the result of the irregularity in distribution of the emphysematous tissue destruction. It is sometimes more clearly recognizable in laminagrams. |
| Lateral—Flattening of the diaphragm | Flattening or even concavity of diaphragmatic contour. A useful index of this change is the presence of a 90-degree or larger sternodiaphragmic angle. In most patients with emphysema, this junction is more readily seen than in subjects with normal chests. |
| Lateral—Abnormal retrosternal space | Abnormal retrosternal space. This is defined as a space showing increased radiolucency and measuring 2.5 cm. Or more from the sternum to the most anterior margin of the ascending aorta. |
| Additional Information | Emphysema is considered to be present if the chest radiolgraphs reveal any two or more of the above criteria. Sometimes it may not be clear not a particular diaphragmic contour is flat. A useful way of resolving this in the posteroanterior radiograph is to determine the straight line from costophrenic junction to the vertebrophrenic junction on each side. If the highest level of the diaphragmic contour is <1.5 cm above this line, the diaphragm may be recorded as flat. The same dimension can be used in the lateral radiograph, measuring from a line connecting the costophrenic junction posteriorly to the sternophrenic junction anteriorly. Flattening of the diaphragmic contours with blunting of costophrenic and sternophrenic angles are seldom, if ever, seen under conditions of acute lung hyperinflation. In addition, areas of irregular radiolucency of the lung fields are absent in such conditions. |

Four signs of Emphysema exactly as described by Sutinen et al. [18]. If the patient has 2 or more of those signs, Sutinen et al. [18] consider this patient as emphysema positive.

## Materials and methods

### Data acquisition

This study was approved by the Institutional Review Board of Radboud University Medical Center (Nijmegen, The Netherlands) (Case number: 2017–3952 Case code: 5PCL). Informed written consent was waived, and data collection and storage were carried out in accordance with local guidelines. The data is made publicly available at https://doi.org/10.5281/zenodo.6373392.

For this retrospective study, we collected 281,000 CXR studies from our hospital archive (2006 to 2019). Only those with a frontal (posteroanterior) and a lateral CXR were retained, resulting in 97,000 studies. Studies where the radiology report mentioned 'emphysema' without the words 'interstitial' or 'subcutaneous' (16,000) were selected for potential inclusion. The dataset was obtained by a random selection of 2000 studies from the potential emphysema studies and 1000 studies from the remaining 81,000 studies. This process is illustrated in Fig 2 and patient statistics are provided.

**Data labeling.** We created two random groups of 1750 studies with 500 studies common to both. Each of these groups was annotated by a chest radiologist, one with over 30 years of experience (ETS) (R1) and the other with over 7 (SS) (R2). Fig 2 illustrates the division of the dataset, with the 500 studies annotated by both radiologists as the test set.

Reader studies on grand-challenge.org [28] were used to annotate images. The radiologists were asked to indicate (yes/no) whether the visual signs of emphysema described by Sutinen

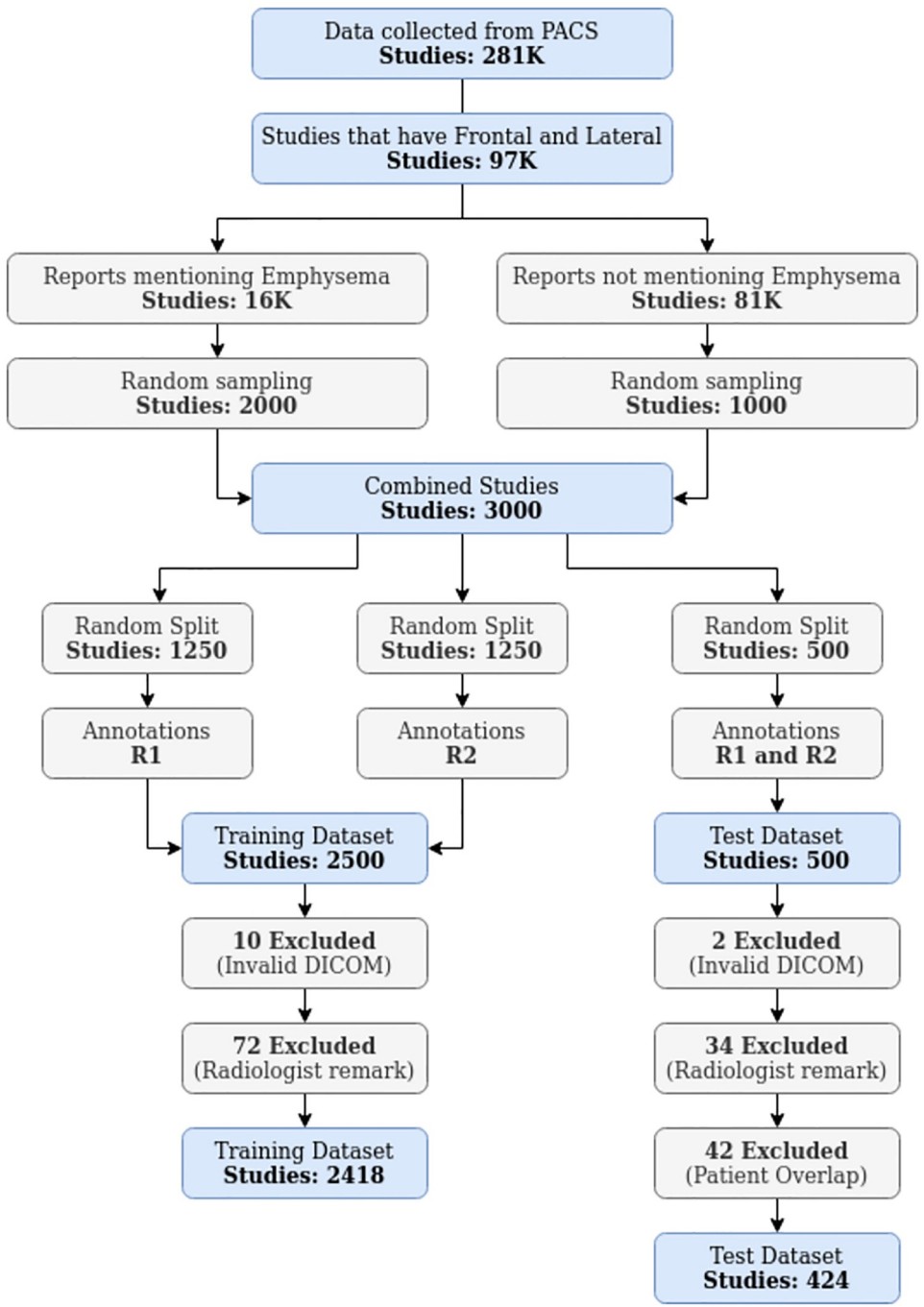

**Fig 2. Data acquisition diagram.**

et al. [18] were present on the images they viewed. The descriptions of these four signs provided to them are reproduced in Table 1. They worked independently, scored frontals and laterals separately, and could not link the frontal and lateral image of a subject with each other. Line drawing and measurement tools were provided, as well as a free-text comment box.

Subjects who had ≥2 of the 4 visual signs indicated present by the radiologist were considered emphysema positive as proposed by Sutinen et al. [18].

For the cases that were annotated by both radiologists (test set), we report the number of positives and negatives for each as well as the Cohen's Kappa and the confusion matrices. For the remaining cases (training set), annotated by either radiologist we report the number of positives and negatives.

## Emphysema sign detection models

To detect the four visual emphysema signs, we trained two ResNet-18 [29] models, one for the frontal and one for the lateral CXRs. ResNet-18 was chosen for this study because it is one of the most frequently used models for deep-learning in CXR [11] and as a relatively shallow model it is suitable for training with a smaller dataset. Each model outputs 2 probabilities indicating scores for the two visual signs of emphysema in the input image. To calculate a final emphysema score for the subject, we average the four probabilities from the two models. The construction and combination of these "sign" models are visualized in Fig 3.

**Preprocessing and data augmentation.** The preprocessing and data augmentation steps are provided in the Supporting Information—see S1 Table in S1 File. Steps 1 to 4 are applied to all images as preprocessing. The data augmentation steps (5–10) are repeated with randomization every time an image is loaded for training (and omitted during validation and testing). The final steps of histogram equalization and resizing (11–12) are applied to all images loaded. These steps were selected heuristically based on initial experimentation with validation data.

**Training.** The training settings of the two sign models are provided in Supporting Information—see S2 Table in S1 File. These were selected heuristically based on experimentation with validation data. We used a multi-stage training procedure, reducing learning rates at each stage and training until the stopping condition was met. At each stage, we loaded the best model from the previous stage, reduced the learning rate, and commenced retraining. The final models are those that achieved the best validation set losses among all stages. The validation set consisting of 256 samples is randomly selected with an equal distribution across labels and annotators and not used for training.

**Ensembling.** Each model is trained 30 times and the resulting probabilities are ensembled using the geometric mean. To increase the variance between models, we used a different held-out validation dataset for each model in the ensemble.

## Black-box model

We trained an ensemble of models that predict emphysema from a frontal and lateral CXR pair without the use of the four visual signs. Similar to the sign models we used the ResNet-18

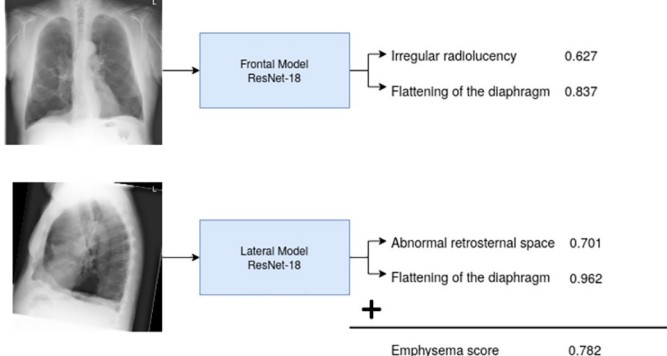

**Fig 3. Illustration of how the emphysema score is created from the sign models.** For each view, we train a separate model predicting the two relevant sign probabilities. In the end, we average these probabilities to calculate a combined emphysema score. The indicated scores are calculated using the CXRs shown in the figure for this specific subject.

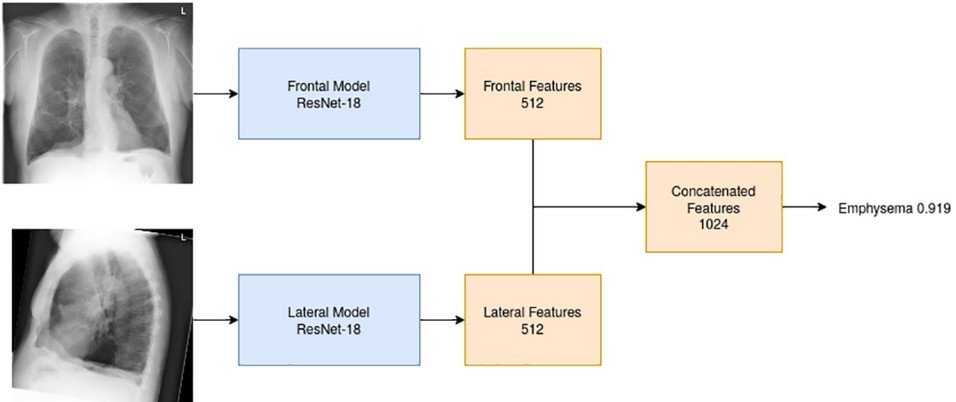

**Fig 4. Illustration of the black-box model.** For each view, we define a model with 512 outputs. The outputs from the two models are concatenated and the probability of emphysema is predicted from this final layer. The indicated score is calculated using the CXRs shown in the figure for this specific subject.

architecture with the same training settings and ensembling process described previously. The frontal and lateral CXR models were combined by concatenating the features from the last feature layers to predict emphysema as shown in Fig 4.

## Model comparison to radiologists

We evaluate the performance of our models using each radiologist separately as the reference standard. Model performances are additionally evaluated on the subset of cases where both radiologists agree (both indicate <2 signs present or both indicate ≥2 signs present for emphysema positivity).

Using the annotations of a single radiologist as the reference standard, the sensitivity and specificity of the other radiologist is calculated. This calculation is repeated for each sign and for emphysema detection by the sign models and by the black-box model. To compare the performance of the radiologist with the models, each model was fixed at the sensitivity of the compared radiologist and the McNemar test [30] was applied to determine a p-value for the performance difference. Statistical significance is inferred if $p < 0.05$. ROC curves with 95% confidence intervals and radiologist sensitivity specificity point with error bars are used for comparison.

This analysis is repeated using the second radiologist as the reference standard, in order to illustrate any model biases that may have been introduced during the annotation and training processes.

## Model comparison to black-box model

We use DeLong's test [31] to evaluate the significance of the performance difference between the black-box model and the emphysema signs model for detecting emphysema. The ROC curves 95% confidence intervals, obtained using bootstrapping, are also provided for comparison. We report the significance of the performance difference for each experiment. As previously, results are provided using each radiologist separately as the reference standard, as well as using only the cases where both radiologists agreed (both indicate <2 signs present or both indicate ≥2 signs present for emphysema positivity).

**Table 2. Patient statistics of the dataset of 3,000 initially selected studies.**

| Gender | Patient count | Patient age | Age range |
|--------|---------------|-------------|-----------|
| **Female** | 1268 | 65.45 ± 12.58 | [22, 99] |
| **Male** | 1732 | 65.53 ± 12.23 | [20, 96] |

## Results

### Data acquisition

Statistics about the 3000 studies are provided in Table 2. During the annotation process, 12 studies were removed because (one of) the DICOM files did not contain valid images, 96 were removed because either the frontal or the lateral were excluded by a radiologist. Exclusion reasons included issues that preclude assessment of the four emphysema signs, such as the diaphragmatic contours not being captured in the chest radiograph, or conditions that hide the signs, such as pleural fluid blocking the view of diaphragmatic contours, or major anatomical changes such as lobectomy.

After exclusions, 2882 studies remained, of which 2418 were annotated by one of the two radiologists (training set), and the remaining 464 (test set) were annotated by both radiologists. Of these 464 studies, 42 were removed to ensure that there was no patient overlap between the training and test datasets, leaving 422 studies that were set aside as the held-out test set.

The number of positive and negative annotations per dataset is provided in Table 3. There were 370 cases (53 positive, 317 negative) in the test set where the radiologists agreed on the emphysema label (both indicate <2 signs present or both indicate ≥2 signs present).

Cohen's kappa scores and confusion matrices for the radiologist agreement on the test dataset are provided in Table 4. All kappa values are within the range of 0.503 and 0.672, indicating that the radiologists had moderate to substantial agreement on all tasks.

### Model performance

The performance of the four individual signs models is presented in terms of AUC in Table 5 and ROC curves are provided in S3-S5 Fig in S1 File. For brevity, the remaining text in this section discusses only the performance of the combined signs model and the black-box model.

**Model comparison to radiologists.** In Fig 5 the ROC curves for the combined signs model and the black-box model are shown using the reference standard from individual radiologists and on the subset of data for which both radiologists agreed. The signs model achieves an AUC of 0.924 or 0.945 depending on the radiologist chosen as the reference standard, or 0.981 on the cases where radiologists agree. Similarly the black-box model has AUCs of 0.915 and 0.935 against the reference standard from single radiologists, or 0.972 on the agreement cases. The performance for these models is also provided in Table 5 along with p-values for accurate comparison. In the comparison with second-reader radiologist we find that neither the signs model nor the black-box model has a performance that is significantly different to an expert radiological reader (Table 5). R2 achieves a sensitivity of 0.779 and specificity of 0.895 (with R1 as the reference standard), while the signs model and the black-box model at the same sensitivity have specificities of 0.901 (p = 0.88) and 0.870 (p = 0.262) respectively. Similarly, using R2 as the reference standard, R1 obtains sensitivity of 0.589 with a specificity of 0.955. At this sensitivity level the signs and black-box model have specificities of 0.985 (p = 0.143) and 0.979 (0.291) respectively.

**Model comparison to black-box model.** For direct comparison of the signs model and the black-box model the ROC curves with 95% confidence intervals are shown in Fig 5 with

**Table 3. Annotation results for the training and test datasets.**

**Training dataset**

| | Patient statistics | |
|---|---|---|
| | Male | Female |
| **Count** | 1384 | 1032 |
| **Age** | 65.7 ± 12.6 | 65.7 ± 12.2 |
| **Age range** | [24, 99] | [20, 95] |
| | **Annotation results** | |
| | Positive | Negative |
| Frontal—Flattening of the diaphragm | 329 | 2089 |
| Frontal—Irregular radiolucency | 318 | 2100 |
| Lateral—Flattening of the diaphragm | 506 | 1912 |
| Lateral—Abnormal retrosternal space | 485 | 1933 |
| Emphysema (≥2 signs) | 425 | 1993 |

**Test dataset**

| | Patient statistics | |
|---|---|---|
| | Male | Female |
| **Count** | 260 | 162 |
| **Age** | 64.4 ± 12.6 | 64.5 ± 12.0 |
| **Age range** | [26, 96] | [37, 93] |

**Test dataset—R1**

| | Annotation results | |
|---|---|---|
| | Positive | Negative |
| Frontal—Flattening of the diaphragm | 40 | 382 |
| Frontal—Irregular radiolucency | 50 | 372 |
| Lateral—Flattening of the diaphragm | 75 | 347 |
| Lateral—Abnormal retrosternal space | 92 | 330 |
| Emphysema (≥2 signs) | 68 | 354 |

**Test dataset—R2**

| | Annotation results | |
|---|---|---|
| | Positive | Negative |
| Frontal—Flattening of the diaphragm | 75 | 347 |
| Frontal—Irregular radiolucency | 63 | 359 |
| Lateral—Flattening of the diaphragm | 88 | 334 |
| Lateral—Abnormal retrosternal space | 95 | 327 |
| Emphysema (≥2 signs) | 90 | 332 |

Note that the test set was annotated by both radiologists independently while the training set was split between them.

AUC values. DeLong's p-values are calculated to compare these. When R1 annotations are taken as the reference standard, there is no significant difference in the AUC values of the signs model and the black box model (AUCs 0.924 and 0.915 respectively) with $p = 0.408$. Similarly, when R2 annotations are considered as the reference standard, there is no significant performance difference (AUC 0.946 for the signs model and AUC of 0.935 for the black box model) with a p-value of 0.345. Performance on the subset of cases where the radiologists agreed also shows no difference between the emphysema sign model (0.981 AUC) and the black-box emphysema model (0.972 AUC) with $p = 0.289$.

**Table 4. Inter-observer variability.**

| Sign and kappa | Confusion matrix | | R2 Neg | R2 Pos |
|---|---|---|---|---|
| Frontal—Flattening of the diaphragm | | | **R2 Neg** | **R2 Pos** |
| | **R1 Neg** | | 340 | 42 |
| Kappa: 0.503 | **R1 Pos** | | 7 | 33 |
| Frontal—Irregular radiolucency | | | **R2 Neg** | **R2 Pos** |
| | **R1 Neg** | | 343 | 29 |
| Kappa: 0.558 | **R1 Pos** | | 16 | 34 |
| Lateral—Flattening of the diaphragm | | | **R2 Neg** | **R2 Pos** |
| | **R1 Neg** | | 316 | 31 |
| Kappa: 0.654 | **R1 Pos** | | 18 | 57 |
| Lateral—Abnormal retrosternal space | | | **R2 Neg** | **R2 Pos** |
| | **R1 Neg** | | 305 | 25 |
| Kappa: 0.672 | **R1 Pos** | | 22 | 70 |
| Emphysema Positive > = 2 positive signs | | | **R2 Neg** | **R2 Pos** |
| | **R1 Neg** | | 317 | 37 |
| Kappa: 0.596 | **R1 Pos** | | 15 | 53 |

Inter-observer variability and confusion matrices of the radiologist annotations for the 4 signs and the emphysema positive result ($\geq$2 positive signs) on 422 cases.

## Discussion

In this retrospective study, we used deep learning on frontal and lateral chest radiographs to detect emphysema using 4 explainable visual signs. Our proposed method, based on these 4 signs of emphysema, performs at the same level as a radiologist (p = 0.880 against R2 and p = 0.143 against R1) in detecting emphysema on CXRs and achieves an AUC of 0.924 or 0.946 against R1 or R2 respectively, or 0.981 on the subset of cases where R1 and R2 agree. We additionally compared our method to a black-box model that did not use explainable visual signs to detect emphysema. Against R1 and R2 this model achieved AUCs of 0.915 and 0.935, while an AUC of 0.972 was obtained on cases where the radiologists agreed. No significant difference was found between the performance of the black-box model and our signs model which has the substantial advantage of providing explainable radiological information.

Emphysema is a condition associated with COPD, which affects 4.6% of the US population [1]. It is relatively difficult to diagnose emphysema conclusively on CXR imaging as evidenced by the moderate kappa scores of the two expert observers involved in this work (Cohen's kappa = 0.596 for $\geq$2 signs present). Previous studies have shown varying sensitivities for the detection of emphysema on CXR. Sanders et al. [32] show a 0.80 sensitivity while Thurlbeck and Simon [33] show sensitivity as low as 0.24. In this work, radiologist R2 had a sensitivity of 0.779 compared to R1. Despite the difficulty of the task, the radiologist is frequently required to identify signs of emphysema on CXR for subjects with suggestive symptoms and history. It is therefore important to be able to consistently and accurately identify such signs to direct patient care appropriately. To our knowledge, this work presents the first deep-learning system focused on emphysema detection in CXR, and is one of very few deep learning systems focusing on explainability of findings in medical image analysis tasks.

The automated diagnosis of emphysema on CXR has received relatively little attention to date. In early work, Coppini et al. [3] used lung boundaries drawn by a physician and specified hand-crafted shape features of these lung boundaries on frontal and lateral CXR. They fed these descriptors into various shallow neural networks to detect emphysema and used the 4

**Table 5. Comparison of the radiologists and the models.**

| Task | R sens. | R spec. | Model spec. at R sens. | p-value | Model AUC |
|---|---|---|---|---|---|
| **Reference Standard R1** | **Compared to R2** | | | | |
| **Models** | | | | | |
| Frontal—Irregular radiolucency | 0.680 | **0.922** | 0.830 | **<0.001** | 0.855 |
| Frontal—Flattening of the diaphragm | 0.825 | 0.890 | **0.939** | **0.004** | 0.949 |
| Lateral—Abnormal retrosternal space | 0.760 | **0.924** | 0.845 | **0.001** | 0.893 |
| Lateral—Flattening of the diaphragm | 0.760 | 0.910 | 0.927 | 0.470 | 0.948 |
| Sign Models Emphysema Positive | 0.779 | 0.895 | 0.901 | 0.880 | 0.924 |
| Black-box Emphysema Positive | 0.779 | 0.895 | 0.870 | 0.262 | 0.915 |
| **Reference Standard R2** | **Compared to R1** | | | | |
| **Models** | | | | | |
| Frontal—Irregular radiolucency | 0.540 | **0.955** | 0.894 | **0.006** | 0.818 |
| Frontal—Flattening of the diaphragm | 0.440 | 0.980 | 0.983 | 1.000 | 0.955 |
| Lateral—Abnormal retrosternal space | 0.737 | 0.933 | 0.917 | 0.635 | 0.923 |
| Lateral—Flattening of the diaphragm | 0.648 | 0.946 | 0.955 | 0.761 | 0.931 |
| Sign Models Emphysema Positive | 0.589 | 0.955 | 0.985 | 0.143 | 0.946 |
| Black-box Emphysema Positive | 0.589 | 0.955 | 0.979 | 0.291 | 0.935 |
| **Cases for which R1 and R2 agree** | | | | | |
| **Models** | | | | | |
| Frontal—Irregular radiolucency | | | | | 0.894 |
| Frontal—Flattening of the diaphragm | | | | | 0.987 |
| Lateral—Abnormal retrosternal space | | | | | 0.941 |
| Lateral—Flattening of the diaphragm | | | | | 0.975 |
| Sign Models Emphysema Positive | | | | | 0.981 |
| Black-box Emphysema Positive | | | | | 0.972 |

Each radiologist is taken as the reference standard, and the performance of the other radiologist and of the model are evaluated. The p-values are obtained using McNemar test [30]. Radiologist sensitivity (R sens.), Radiologist specificity (R spec.), the model specificity at Radiologist sensitivity (Model spec. at R sens.) and the Model AUC are provided. Bold p-values indicate $p < 0.05$, and bold specificity values indicate significantly higher specificity in the given comparison. The last section of the table provides AUC values for the models evaluated on the subset of cases where R1 and R2 agreed on emphysema positivity.

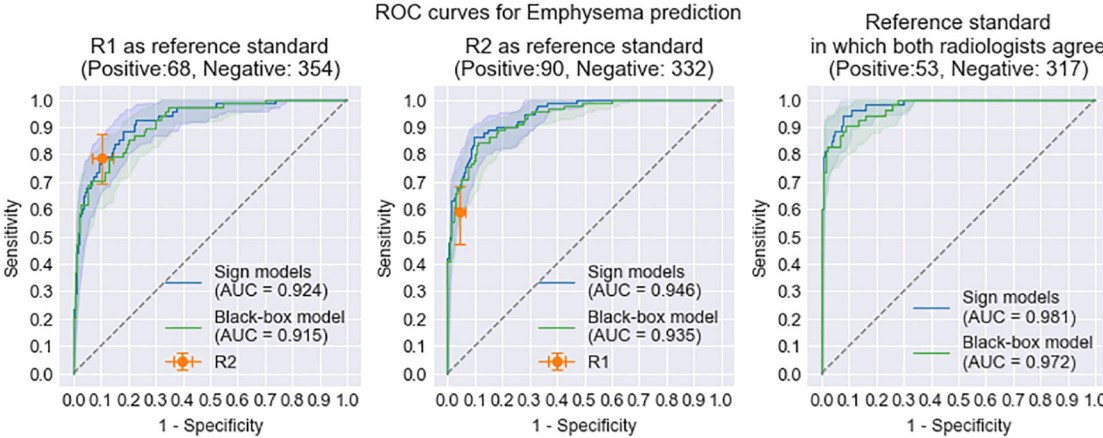

**Fig 5. ROC curve comparison of the combined emphysema signs model for detecting emphysema with at least 2 signs, the black-box emphysema model, and a radiologists sensitivity specificity point.** ROC curves are drawn for 2 different reference standards. R1 as reference standard (left)and R2 as reference standard (centre), and finally for only those cases where the radiologists agreed on the emphysema label (right). The 95% confidence intervals and error bars are calculated by bootstrapping.

signs from Sutinen et al. [18] to label their dataset. Their study used a dataset of just 320 studies with 60 emphysema positives and obtained 0.90 accuracies using 10 fold cross-validation. In their follow-up work, Miniati et al. [5] similarly collected frontal and lateral lung segmentations of 225 studies from a physician. They had 92 emphysema subjects and split their dataset into training (118) and validation (107) sets. This work used CT confirmed emphysema labels and again, hand-crafted features that describe the lung shapes. Using a shallow neural network to obtain an emphysema classification, they achieved an AUC of 0.955. In another follow-up study, Coppini et al. [4] automated the segmentation of lung boundaries and achieved 0.954 AUC for the same task on the same dataset. These works use small datasets and appear to optimize their experimental results on the test sets, meaning that they are unlikely to generalize to large unseen datasets. More recently, Campo et al. [7] simulated chest radiographs from CT scans. From these CT scans, they automatically generated the percentage of low-attenuation lung areas (%LLA, Müller et al. [34]) to determine the ratio of emphysematous tissue volume. Using this reference standard, they experiment with various %LLA thresholds to define emphysema and train CNNs consisting of 11 layers (4 convolutional). Using a 10%LLA threshold, on a dataset of 2666 training and 4671 test samples, they achieved 0.907 AUC in classifying emphysema in simulated CXRs. This work demonstrates the potential to detect emphysema automatically on CXR but does not use any real CXR images and the results cannot be considered generalizable to that domain. Li et al. [16] annotated 10,738 studies by 3 radiologists for 14 disease labels of ChestXray14 and reported 0.943 AUC in predicting emphysema. One drawback of this approach is that the agreement of 3 radiologists is likely to be favoring more severe cases of emphysema as positive.

Many studies use the ChestXray14 dataset [10], which contains the emphysema label obtained by automatically parsing radiology reports. One recent example Wang et al. (2021) [14] proposed an extension of DenseNet121 [15] using various attention modules and achieved 0.933 AUC in detecting emphysema, which is comparable to our method. However, we note that the emphysema labels in this dataset are unreliable, having seen in our previous work [12], that 39 of 90 randomly selected emphysema samples had incorrect labels. Oakden-Rayner [13] demonstrated the same issue, finding that 86% of visually examined emphysema cases from that dataset were, in fact, cases of subcutaneous emphysema rather than pulmonary emphysema. This issue casts doubt on the emphysema classification performance of the many similar studies that use the ChestXray14 dataset labels for evaluation [11] and so we omit any direct comparison with these works.

The work of Miniati et al. [19] demonstrated that the number of positive signs on the CXR image correlated with the severity of emphysema on CT. While we are unable to demonstrate such a finding without reference standard severity scores, it seems likely that the number of positive signs, or indeed the scores assigned by the deep learning systems would correlate with disease severity. This is an interesting avenue for future research.

One limitation of this work is the lack of reference standard for the emphysema label based on CT imaging or confirmed emphysema diagnosis. The models presented here are trained only to emulate the performance of a radiologist identifying emphysema on a chest X-Ray, and are evaluated in that context also. This does not provide any indication of how well CXR-based analysis compares with more accurate reference standards such as quantitative CT and/or clinical diagnosis. Future studies should endeavor to obtain such data for a more accurate analysis of performance. The inclusion of additional expert reader opinions and data from a different institution would also improve the analysis in future work. Finally, a more systematic search of the parameter space may identify improved settings for the deep learning systems used in this work.

## Conclusion

This work presents the first fully automatic and explainable deep-learning system for the detection of emphysema on CXR. Using a large and manually-labeled dataset with held-out test data from 422 studies, we demonstrate that the proposed method has a performance equivalent to an expert radiologist and to a black-box system that provides no explainable features. This work demonstrates the feasibility of providing explainable features through deep-learning systems as well as a potentially useful tool for emphysema detection.

## Supporting information

**S1 File. Supporting information file.** Collection of all supporting tables and figures. Including S1, S2 Tables, S3-S5 Figs.
(PDF)

## Author Contributions

**Conceptualization:** Erdi Çallı, Keelin Murphy, Bram van Ginneken.

**Data curation:** Erdi Çallı, Ernst T. Scholten, Steven Schalekamp.

**Formal analysis:** Erdi Çallı.

**Funding acquisition:** Bram van Ginneken.

**Investigation:** Erdi Çallı.

**Methodology:** Erdi Çallı, Keelin Murphy, Ernst T. Scholten, Bram van Ginneken.

**Software:** Erdi Çallı.

**Supervision:** Keelin Murphy, Bram van Ginneken.

**Validation:** Erdi Çallı.

**Visualization:** Erdi Çallı.

**Writing – original draft:** Erdi Çallı.

**Writing – review & editing:** Keelin Murphy, Bram van Ginneken.

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
