## [Decision Letter · Decision Letter 0]

8 Feb 2022

PONE-D-21-36550Explainable emphysema detection on chest radiographs with deep learningPLOS ONE

Dear Dr. Çallı,

Thank you for submitting your manuscript to PLOS ONE. After careful consideration, we feel that it has merit but does not fully meet PLOS ONE’s publication criteria as it currently stands. Therefore, we invite you to submit a revised version of the manuscript that addresses the points raised during the review process.

 Please revise your article according to each of the suggestions given by the reviewers especially in the design of experiments. 

We look forward to receiving your revised manuscript.

Kind regards,

Yan Chai Hum

Academic Editor

PLOS ONE

Journal Requirements:

2. Please update your submission to use the PLOS LaTeX template. The template and more information on our requirements for LaTeX submissions can be found at 

Reviewers' comments:

Reviewer's Responses to Questions

**Comments to the Author**

1. Is the manuscript technically sound, and do the data support the conclusions?

Reviewer #1: Partly

Reviewer #2: Yes

Reviewer #3: Yes

2. Has the statistical analysis been performed appropriately and rigorously? 

Reviewer #1: No

Reviewer #2: Yes

Reviewer #3: Yes

3. Have the authors made all data underlying the findings in their manuscript fully available?

Reviewer #1: Yes

Reviewer #2: Yes

Reviewer #3: Yes

4. Is the manuscript presented in an intelligible fashion and written in standard English?

Reviewer #1: Yes

Reviewer #2: Yes

Reviewer #3: Yes

5. Review Comments to the Author

Reviewer #1: In this research, authors apply DL algos on emphysema detection. The research is interesting, however, the proposed methodology existed in the literature and thus the novelty of the research is a concern to be published in PLOS ONE. So I suggest to give the author a chance to submit after doing additional research to improve the contributions of the research work. Authors may also consider the following comments for the revision work:

1. For the literature review, authors should refer to more recent research, i.e. year 2020-2021. Currently there is no references for year 2021. Authors need to refer to more ISI/Scopus research work instead of the online/conference resources. Currently, there is only 21 references which can be still improve for a ISI level research paper.

2. Authors need to identify the research gap and highlight the contribution(s) of the research work in the manuscript to shows the novelty of the research.

3. Justification is needed for all the selected parameters, i.e. why the technique is being selected instead of other existing techniques?

4. More scientific reasoning should be added in the experimental results' explanations.

5. The format of the manuscript needs to be improved. Some sections need to be combined and restructured.

6. The results of the proposed methods should be compare with the state-of-the-arts methods in order to shows the novelty / contributions of the research work.

7. Only a self-annotated dataset is not convincing enough. Authors should include an open source dataset to prove that the model is general enough and the results and discussion would then be more convincing.

Reviewer #2: Summary

The authors propose to diagnose lung emphysema from a pair of frontal-lateral chest x-rays. The main contribution is in departing from direct classification of the image pair in favor of detecting the presence of four radiological markers, and deriving an "emphysema score" from marker predictions. The experiments demonstrate the feasibility of this approach, but the difference between a direct "black box" approach and the proposed method is not statistically significant.

Minor weakness

The design of the experimental evaluation makes it very difficult to interpret the results.

Each of the test radiographs received annotations from two radiologists in terms of presence of four markers. For three out of these four markers, the doctors disagreed almost as often as (or more often than) they agreed that the marker is present (Tab. 4).

The authors trained a deep net on equal number of annotations from both radiologists, but computed test scores using annotations from one of them only. I find it difficult to interpret these scores.

More precisely, a hypothetical upper bound on performance of a system trained and tested on annotations from the same distribution is determined by the variance of the annotations. In case of noise-free annotations, a "perfectly accurate" system could operate with precision and recall =1. But labels produced by two doctors likely come from two different distributions (as suggested by Tab 4). A "perfectly accurate" deep network, trained on such mixture of labels, would learn to fit the mixture of the two distributions. When evaluated against labels originating from one of the distributions, even the "perfectly accurate" network should not be expected to attain maximum test scores.

This is exactly the case of the presented experiments. I am unable to interpret the "specificity" of a network trained to fit the mixture of two distributions in reproducing samples of one of its components (Tab. 5). Moreover, the comparison to the other doctor (R2) cannot be interpreted in terms of "comparison to human performance", as implied by the authors. Such a claim would only make sense if both the network and the doctor were tasked with predicting results of some objective test, like the lung function test, or the clinical outcome. If the system was trained on annotations performed by the first doctor (R1), then the authors could at least compare the disagreement between the system and R1 to the disagreement between R1 and R2. In the current setup the numerical results are difficult to interpret, because the system is not trained to agree with R1, but to interpolate between R1 and R2. For example, it is not clear what it means that the proposed method is "worse than R2" on detecting two of the markers (Tab. 5). How far is it from the mixture of the distributions of R1 and R2, that it was trained to approximate?

To address this difficulty, I suggest that the authors extend the experimental evaluation, and the associated description, in one of two ways:

1. Either add an evaluation limited to the test scans on which the two radiologists agreed,

2. or train the network on annotations of one of the radiologists, then compare the disagreement between that radiologist and the system to the disagreement between the two radiologists.

Justification of the rating

The work appears to be methodologically correct. I suggest extending the description of the experimental evaluation according to my detailed comment above, to facilitate interpretation of numerical performance of the proposed system. It should be straightforward for the authors to add this additional result to the final version of the manuscript, which does not require further review.

Editorial comment

Please use consistent terminology across the manuscript: either "test set" or "evaluation set". Around lines 84 and 89, please state explicitly that the training set is split into the "actual" training set and a small validation set.

Reviewer #3: In this study a deep learning system is proposed to detect emphysema on chest radiographs. It is shown that the proposed method is able to predict emphysema positivity with a performance that is comparable to that of a radiologist. The paper is well written and there are only minor issues to be addressed so that it could be recommended for publication.

My only main critical note to the study is an issue that the authors themselves hint to in the Discussion section: the labeling of the 4 visual signs related to emphysema by a single radiologist are taken as ground truth and this can potentially cause some bias to the models introduced and the results obtained with them. The confusion matrices of radiologist annotations (Table 4) and kappa values indicate that there are a significant number of cases, where the two radiologist do not agree on positivity compared to the number of those where both of them signaled positivity for a given visual sign. There are a couple of questions here that in my opinion the authors would need to address:

- What is the confusion matrix for overall emphysema positivity (taken after the rule of at least 2 positive signs)? This has been not indicated in the paper.

- Given the moderate disagreement between the radiologists, why only R1-annotated radiographs were taken as ground truth and no evaluation has been done considering R2-annotated radiographs as ground truth and/or taking only those graphs as such where there were an agreement between R1 and R2? Such an analysis would shed more light to any potential bias present in the models.

- What about availability of actually diagnosed emphysema with corresponding radiographs? The only truly unbiased assessment can be made only in the case when ground truth for diagnosis is taken independently from the visual signs on radiographs.

Other minor points:

- Fig. 3 and 4: In the captions please indicate that the displayed probabilities correspond to a specific study case.

- Table 4: In the upper row the legend of 'R1 Neg' is missing to the confusion matrix

6. PLOS authors have the option to publish the peer review history of their article (what does this mean?). If published, this will include your full peer review and any attached files.

Reviewer #1: No

Reviewer #2: No

Reviewer #3: No

---

## [Author Response · Author response to Decision Letter 0]

20 Mar 2022

Dear Editor, 

Thank you for taking the time to arrange the review of our manuscript. In the following document, we address all the reviewer comments. Thanks to the feedback, we are confident that our manuscript has improved. 

For each reviewer, we define a section (such as Reviewer #1) and for each comment of each reviewer, we define a subsection (such as R1-C1 for the first comment of the first reviewer). We define a feedback subsection for not actionable feedback (such as R1-Feedback). Under each of these subsections, we provide the reviewer's comment as well as our response.

We are looking forward to hearing your decision.

Kind regards,

Erdi Calli

Reviewer #1 

R1-Feedback

Reviewer: In this research, authors apply DL algos on emphysema detection. The research is interesting, however, the proposed methodology existed in the literature and thus the novelty of the research is a concern to be published in PLOS ONE. So I suggest giving the author a chance to submit after doing additional research to improve the contributions of the research work. Authors may also consider the following comments for the revision work:

Response from authors: Dear Reviewer, thank you very much for your insights and comments on our research. Following your suggestions, we have greatly changed our study. Please see the 7 responses that we provided corresponding to the 7 comments that you have written. 

R1-C1

Reviewer: For the literature review, authors should refer to more recent research, i.e. year 2020-2021. Currently, there are no references for the year 2021. Authors need to refer to more ISI/Scopus research work instead of the online/conference resources. Currently, there are only 21 references that can be still improved for an ISI-level research paper.

Response from authors: Based on this feedback we performed a search on the peer-reviewed publications using the following query (chest and (“x-ray” or “radiograph”) and emphysema and (“machine learning” or “deep learning” or “artificial intelligence”). This search led us to add 3 new studies into our publication that have either been published after we submitted the paper, or published after we conducted our initial search. Unfortunately, emphysema detection with deep learning is a field that has not yet attracted a lot of attention. Also, the publicly available label data for this disease is extremely noisy as we explain in lines 271-282. These properties make it difficult to find studies for comparison since the emphysema labels used in most works are extremely unreliable. We mentioned these details in our introduction and discussion in detail (lines 27-40, 271-282). We have also noted other locations throughout the text where citations could be included to improve the manuscript during the review process and have added these increasing the total number of references from 21 to 36. In the following list, you can see which reference is added as a response to which comment (such as R1-C1 for the current comment), the authors with the year (as Huang et al. (2017)), followed by the lines in which this citation is used.

R1-C6: Huang et al. (2017), 31-35, 272-276

R1-C2: Kashyap et al. (2020), 50-59

R1-C2: Lee et al. (2021), 50-59

R1-C1: Li et al. (2021), 35-37, 267-270

R1-C1: Lin et al. (2020), 37-40

R1-C2: Nguyen et al. (2019), 50-59

R1-C2: Pasa et al. (2019), 50-59

R1-C1: Pu et al. (2021), 21-24

R1-C2: Rajpurkar et al. (2017), 50-59

R1-C2: Ras et al. (2021), 50-59

R1-C2: Sogancioglu et al. (2020), 50-59

R1-C1: Urban et al. (2022), 24-26

R1-C1: Wanchaitanawong et al. (2021),15-17

R1-C6: Wang et al. (2021), 31-34, 272-276

R1-C2: Xie et al. (2020), 50-59

R1-C2

Reviewer: Authors need to identify the research gap and highlight the contribution(s) of the research work in the manuscript to show the novelty of the research.

Response from authors: We agree that it is important to further highlight our contribution with this work. To identify the research gap, highlight our contribution, and provide the novelty of our application, we have added a paragraph in the introductions (lines 50-59). We mentioned the need for an explainable system that can be integrated into the radiology workflow and that is easily interpreted by the users in this domain.

R1-C3

Reviewer: Justification is needed for all the selected parameters, i.e. why the technique is being selected instead of other existing techniques?

Response from authors: We thank the reviewer for this suggestion. The existing text mentions that our training parameters were selected based on initial experimentation on validation data. We have added a further explanation on why we choose Resnet18 as the most optimal model and additionally explained that other parameters were chosen heuristically based on initial experiments.(lines 105-107, 117-118).

R1-C4

Reviewer: More scientific reasoning should be added in the experimental results' explanations.

Response from authors: We have expanded our discussion section with a more thorough explanation of results, referring to the p-values for scientific reasoning. (lines 221-228).

R1-C5

Reviewer: The format of the manuscript needs to be improved. Some sections need to be combined and restructured.

Response from authors: We identified some restructuring steps that would improve the format of the manuscript and we thank the reviewer for this suggestion. We have divided the methods section into more subsections and added the “Model comparison to radiologists” and “Model comparison to black-box model” subsections. Each subsection explains one portion of the work. These subsections are reflected with the same titles in the Results section. Please see lines 139-163, and 186-217 for these changes. We also tried to increase consistency in various places such as figures, figure captions and improved the text in various places. Since these changes are widespread throughout the manuscript we do not include every change in this document as it is not straightforward to follow them in this way.

R1-C6

Reviewer: The results of the proposed methods should be compared with the state-of-the-art methods in order to show the novelty/contributions of the research work.

Response from authors: We agree with the reviewer that comparison with state of the art is very important, and while we have done our best to draw comparisons this is difficult given that no other work to date has used the same set of data or aimed for precisely the same task, and that many works use public datasets with inaccurate labels. In our Discussion section, we mention various related studies and compare the task, AUC, and data used (see lines 243-282). We have added an extra comparison based on the literature findings of R1-C1 and compared our method to Li et al. (2021) (lines 267-270). We additionally have added specific mention of a recent study which to our knowledge, has achieved one of the best results so far. “Triple attention learning for classification of 14 thoracic diseases using chest radiography”, Wang et al. (2021). We mention the result of this study (AUC 0.933 for emphysema detection) in our discussions and mention how it compares to our method (lines 272-274). However, we must observe, as with most previous works, that this study uses public data with automatically extracted emphysema labels which cannot be considered reliable. We detail these in the Discussion section (lines 275-282).

R1-C7

Reviewer: Only a self-annotated dataset is not convincing enough. Authors should include an open-source dataset to prove that the model is general enough and the results and discussion would then be more convincing.

Response from authors: We appreciate this suggestion from the reviewer and we certainly agree that an additional dataset from an external institution or an open dataset would be a great contribution to the paper. This was discussed during the work for this study but unfortunately, we came to the conclusion that this data should be prepared for a follow-up study because of the difficulties of collecting such a dataset. To be able to do this task correctly, we need to collect a hand-labeled dataset, which requires a great deal of time from one or more radiologists. This is necessary because it is known that emphysema labels in the public CXR datasets (which are extracted automatically from radiology reports and contain many inaccuracies) are unsuitable for immediate use (see lines 27-31, 275-282 of our introduction and discussion). To obtain a meaningful number of positive samples, a large amount of data has to be reviewed and labeled by radiologists and unfortunately, it is very difficult for our radiologists to spare time for these tasks. To label, the 3000 images included in this study took approximately one year since the readers were frequently unable to attend to the labeling based on their workload. Radiologist time is very costly and scarce but we believe it is essential for the creation of a high-quality dataset. Furthermore, we note that most public CXR datasets do not include lateral images, which are required for this work, and correctly extracting and checking frontal and lateral pairs and cleaning the dataset prior to labeling is also a substantial amount of work. We regret that we are unable to include additional datasets at this time but we have identified this suggestion of the reviewer as a priority task for future development and validation of our method. We note that this is also described in the limitations of the study (lines 295-297) as shown below.

R1-Summary

Authors: Again, thanks a lot for your well-thought-out and detailed instructions on the necessary changes. We believe that answering your comments improved our paper.

Reviewer #2

R2-Feedback

Reviewer: The authors propose to diagnose lung emphysema from a pair of frontal-lateral chest x-rays. The main contribution is in departing from direct classification of the image pair in favor of detecting the presence of four radiological markers, and deriving an "emphysema score" from marker predictions. The experiments demonstrate the feasibility of this approach, but the difference between a direct "black box" approach and the proposed method is not statistically significant.

Response from authors: Dear reviewer, Thank you very much for reading our manuscript and providing valuable feedback. 

R2-C1

Reviewer: Minor weakness

The design of the experimental evaluation makes it very difficult to interpret the results.

Each of the test radiographs received annotations from two radiologists in terms of the presence of four markers. For three out of these four markers, the doctors disagreed almost as often as (or more often than) they agreed that the marker is present (Tab. 4).

The authors trained a deep net on an equal number of annotations from both radiologists, but computed test scores using annotations from one of them only. I find it difficult to interpret these scores.

More precisely, a hypothetical upper bound on the performance of a system trained and tested on annotations from the same distribution is determined by the variance of the annotations. In the case of noise-free annotations, a "perfectly accurate" system could operate with precision and recall =1. But labels produced by two doctors likely come from two different distributions (as suggested by Tab 4). A "perfectly accurate" deep network, trained on such a mixture of labels, would learn to fit the mixture of the two distributions. When evaluated against labels originating from one of the distributions, even the "perfectly accurate" network should not be expected to attain maximum test scores.

This is exactly the case of the presented experiments. I am unable to interpret the "specificity" of a network trained to fit the mixture of two distributions in reproducing samples of one of its components (Tab. 5). Moreover, the comparison to the other doctor (R2) cannot be interpreted in terms of "comparison to human performance", as implied by the authors. Such a claim would only make sense if both the network and the doctor were tasked with predicting the results of some objective test, like the lung function test, or the clinical outcome. If the system was trained on annotations performed by the first doctor (R1), then the authors could at least compare the disagreement between the system and R1 to the disagreement between R1 and R2. In the current setup the numerical results are difficult to interpret because the system is not trained to agree with R1, but to interpolate between R1 and R2. For example, it is not clear what it means that the proposed method is "worse than R2" on detecting two of the markers (Tab. 5). How far is it from the mixture of the distributions of R1 and R2, that it was trained to approximate?

To address this difficulty, I suggest that the authors extend the experimental evaluation, and the associated description, in one of two ways:

Either add an evaluation limited to the test scans on which the two radiologists agreed,

or train the network on annotations of one of the radiologists, then compare the disagreement between that radiologist and the system to the disagreement between the two radiologists.

Response from authors: Thank you very much for your detailed explanation. We understand and agree with the difficulty of interpreting the results. These two are excellent suggestions. Doing the second item would halve the number of training samples and have a big impact on the results, so we opted for the first item. Based on your feedback, we have extended our experiment design and included two new sets of results. Firstly, we included the results where the R2 is taken as the ground truth. This change can be seen in Table 5, Figure 5, and Supplementary information S4 Fig, as well as S5 Fig. The necessity for extra ROC curves has led us to move some figures into the supplementary materials for better readability and add the AUC values for those in Table 5 for completeness within the main text. Secondly, we included results for the agreed-upon annotations (subjects where both radiologists agreed on the emphysema diagnosis) to give a sense of the performance of the method. 

R2-Feedback

Reviewer: Justification of the rating

The work appears to be methodologically correct. I suggest extending the description of the experimental evaluation according to my detailed comment above, to facilitate interpretation of numerical performance of the proposed system. It should be straightforward for the authors to add this additional result to the final version of the manuscript, which does not require further review.

Response from authors: Thank you very much for sharing your observations and very detailed feedback. 

R2-Q2

Reviewer: Editorial comment

Please use consistent terminology across the manuscript: either "test set" or "evaluation set". 

Response from authors: We have made the necessary changes through the manuscript for consistency.

R2-Q3

Reviewer: Around lines 84 and 89, please state explicitly that the training set is split into the "actual" training set and a small validation set.

Response from authors: Our experiment design did not include a pre-specified held-out validation dataset. However, everytime we trained our models, we made sure to hold-out a random part of the training dataset for validation. This was mentioned in the Ensembling section. We moved this definition under the training section (lines 126-127), and added a mention of held-out in line 127. 

R2-Summary

Authors: We would like to thank you again for your detailed feedback. We think that the manuscript has improved greatly based on it. 

Reviewer #3

R3-Feedback

Reviewer: In this study, a deep learning system is proposed to detect emphysema on chest radiographs. It is shown that the proposed method is able to predict emphysema positivity with a performance that is comparable to that of a radiologist. The paper is well written and there are only minor issues to be addressed so that it could be recommended for publication.

Response from authors: Dear reviewer, thank you very much for reading our manuscript and providing valuable feedback. 

R3-Feedback

Reviewer: My only main critical note to the study is an issue that the authors themselves hint at in the Discussion section: the labeling of the 4 visual signs related to emphysema by a single radiologist is taken as ground truth and this can potentially cause some bias to the models introduced and the results obtained with them. The confusion matrices of radiologist annotations (Table 4) and kappa values indicate that there are a significant number of cases, where the two radiologists do not agree on positivity compared to the number of those where both of them signaled positivity for a given visual sign. There are a couple of questions here that in my opinion, the authors would need to address:

Response from authors: We agree on the necessity of clarifying these questions and believe these have been addressed in the subsequent responses to the reviewer (R3-C1, R3-Q2, and R3-Q3).

R3-C1

Reviewer: What is the confusion matrix for overall emphysema positivity (taken after the rule of at least 2 positive signs)? This has been not indicated in the paper.

Response from authors: The confusion matrix for overall emphysema positivity is provided in Table 4, please see the final row of the table for the emphysema confusion matrix as shown in the following image.

R3-Q2

Reviewer: Given the moderate disagreement between the radiologists, why only R1-annotated radiographs were taken as ground truth and no evaluation has been done considering R2-annotated radiographs as ground truth and/or taking only those graphs as such where there was an agreement between R1 and R2? Such an analysis would shed more light on any potential bias present in the models.

Response from authors: We understand and agree with the difficulty of interpreting the results. Another reviewer also made the same point about using the annotations from the other radiologist. Based on these answers we have extended our experiment design. We included the results where the R2 is taken as the ground truth and results for the agreed-upon annotations to give a better sense of the performance of the method. Please refer to the response we provided in section R2-C1 of this document for further information on this item. 

R3-Q3

Reviewer: What about the availability of actually diagnosed emphysema with corresponding radiographs? The only truly unbiased assessment can be made only in the case when ground truth for diagnosis is taken independently from the visual signs on radiographs.

Response from authors: We thank the reviewer for raising this point and we agree that this is an important issue. It would be ideal if the data included a CT or clinical diagnosis as the reference standard. Unfortunately, since this data was not available to us, we aimed to produce a model which could perform at a similar level to a radiologist viewing a chest X-Ray, and we evaluate our model only in that context. To determine whether viewing the chest X-Ray is sufficient to provide accurate emphysema diagnosis, we would certainly need to include additional radiological or clinical data. We tried to address this in the discussion section (lines 290-294), and have now extended this text for clarity. 

R3-Q4

Reviewer: Other minor points: Fig. 3 and 4: In the captions please indicate that the displayed probabilities correspond to a specific study case.

Response from authors: Fig. 3 and 4 captions are changed to include this information.

R3-Q5

Reviewer: Table 4: In the upper row the legend of 'R1 Neg' is missing from the confusion matrix

Response from authors: We have added the missing “R1 Neg” in Table 4.

R3-Summary

Authors: We would like to thank you for your feedback. We think that the manuscript has improved greatly based on your feedback. 

Journal Requirements

Requirement-1

Editor: When submitting your revision, we need you to address these additional requirements.

Response from authors: The article suits the PLOS style requirements.

Requirement-2

Editor: Please update your submission to use the PLOS LaTeX template. The template and more information on our requirements for LaTeX submissions can be found at 

Response from authors: The article uses the PLOS template.

Requirement-3

Editor: We note that you have stated that you will provide repository information for your data at acceptance. Should your manuscript be accepted for publication, we will hold it until you provide the relevant accession numbers or DOIs necessary to access your data. If you wish to make changes to your Data Availability statement, please describe these changes in your cover letter and we will update your Data Availability statement to reflect the information you provide.

Response from authors: Upon acceptance, the data will be made publicly available and the link will be added to the manuscript.

Requirement-4

Editor: Please include your full ethics statement in the ‘Methods’ section of your manuscript file. In your statement, please include the full name of the IRB or ethics committee who approved or waived your study, as well as whether or not you obtained informed written or verbal consent. If consent was waived for your study, please include this information in your statement as well. 

Response from authors: Full ethics statement (The study has been reviewed by the ethics committee on the basis of the Dutch Code of conduct for health research, the Dutch Code of conduct for responsible use, the Dutch Personal Data Protection Act and the Medical Treatment Agreement Act (CMO: 2017-3952).) is added, please see lines 74-77.

Requirement-5

Editor: Please include captions for your Supporting Information files at the end of your manuscript, and update any in-text citations to match accordingly. Please see our Supporting Information guidelines for more information: http://journals.plos.org/plosone/s/supporting-information. 

Response from authors: Captions for supporting information are included in the initial manuscript. For new supporting information, captions are added.

---

## [Editor Report · Decision Letter 1]

12 Apr 2022

Explainable emphysema detection on chest radiographs with deep learning

PONE-D-21-36550R1

Dear Dr. Çallı,

We’re pleased to inform you that your manuscript has been judged scientifically suitable for publication and will be formally accepted for publication once it meets all outstanding technical requirements.

Kind regards,

Yan Chai Hum

Academic Editor

PLOS ONE

Additional Editor Comments (optional):

All concerns have been addressed.
---

## [Editor Report · Acceptance letter]

22 Jun 2022

PONE-D-21-36550R1 

Explainable emphysema detection on chest radiographs with deep learning 

Dear Dr. Çallı:

I'm pleased to inform you that your manuscript has been deemed suitable for publication in PLOS ONE. Congratulations! Your manuscript is now with our production department. 

Kind regards, 

on behalf of

Dr. Yan Chai Hum 

Academic Editor

PLOS ONE